# Differential Tropism in Roots and Shoots of Resistant and Susceptible Cassava (*Manihot esculenta* Crantz) Infected by Cassava Brown Streak Viruses

**DOI:** 10.3390/cells10051221

**Published:** 2021-05-17

**Authors:** Samar Sheat, Paolo Margaria, Stephan Winter

**Affiliations:** Plant Virus Department, Leibniz Institute DSMZ-German Collection of Microorganisms and Cell Cultures, 38124 Braunschweig, Germany; Samar.Sheat@dsmz.de (S.S.); Paolo.Margaria@dsmz.de (P.M.)

**Keywords:** *Ipomovirus*, virus resistance, phloem unloading, companion cells, tissue localization, cryo-sectioning, RNAscope^®^ ISH

## Abstract

Cassava brown streak disease (CBSD) is a destructive disease of cassava in Eastern and Central Africa. Because there was no source of resistance in African varieties to provide complete protection against the viruses causing the disease, we searched in South American germplasm and identified cassava lines that did not become infected with the cassava brown streak viruses. These findings motivated further investigations into the mechanism of virus resistance. We used RNAscope^®^ in situ hybridization to localize cassava brown streak virus in cassava germplasm lines that were highly resistant (DSC 167, immune) or that restricted virus infections to stems and roots only (DSC 260). We show that the resistance in those lines is not a restriction of long-distance movement but due to preventing virus unloading from the phloem into parenchyma cells for replication, thus restricting the virus to the phloem cells only. When DSC 167 and DSC 260 were compared for virus invasion, only a low CBSV signal was found in phloem tissue of DSC 167, indicating that there is no replication in this host, while the presence of intense hybridization signals in the phloem of DSC 260 provided evidence for virus replication in companion cells. In neither of the two lines studied was there evidence of virus replication outside the phloem tissues. Thus, we conclude that in resistant cassava lines, CBSV is confined to the phloem tissues only, in which virus replication can still take place or is arrested.

## 1. Introduction

Cassava brown streak disease (CBSD) is the most devastating disease of cassava (*Manihot esculenta* Crantz) that strongly affects the quality and yield of this important food crop’s storage roots. Tuberous roots of virus-infected plants show brownish necrotic areas varying in extent, from a few corky patches visible only on a few tubers to extensive necrosis resulting in the total destruction of the tuberous roots. The disease is caused by cassava brown streak virus (CBSV) and Uganda cassava brown streak virus (U/CBSV), both being distinct species of the genus *Ipomovirus* in the family *Potyviridae* [1]. The two viruses (U/CBSV) cause indiscriminate symptoms [1,2,3] in cassava but differ in plant invasion speed and the degree of replication. CBSV is considered the more aggressive virus, quickly moving through the infected plant and reaching high virus concentrations in most cassava varieties tested [4,5].

Intensive efforts have been made to identify resistance sources against U/CBSV [6,7] but, so far, high resistance in African varieties has not been found. The cassava varieties considered resistant have lower virus titers and reduced symptom expression in leaves and roots [2,3,8]; however, all genotypes so far tested can still become virus infected and eventually show root symptoms. We recently identified cassava genotypes in South American cassava germplasm maintained in the CIAT (International Center for Tropical Agriculture, Columbia) collection that show high resistance against U/CBSV [4]. We identified several lines immune to CBSV and we found others in which the virus infection was confined to the roots and no virus was detectable in leaf tissues.

In the “immune” cassava, accession DSC 167 (CIAT Col 2182), we previously found no evidence for virus replication and neither of the two viruses U/CBSV were traceable in leaves or tuberous roots [4]. This observation led us to speculate on the fate of the virus in a resistant line, in particular when the virus is continuously delivered from an infected susceptible source through a vascular tissue junction established by grafting. Similarly, the observation of virus invasion and replication in root tissues, but the restriction of the virus to roots and stems in the accession DSC 260 (CIAT PER 353), raised questions about the presence of physical barriers impeding cell-to-cell and/or long-distance movement and the assortment of resistance mechanisms that may be activated to prevent virus replication and plant invasion.

Plants utilize diverse defense mechanisms to restrict virus replication and accumulation in infected tissues as well as to counteract viral cell-to-cell or long-distance movement [9,10]. Gene silencing, immune receptor signaling, protein degradation, regulation of metabolism and hormone-mediated defenses are all defense pathways that the virus has to face to establish infection [11,12,13]. However, prior to biochemical and molecular studies that would lead to a mechanistic description of resistance, we were interested to further study virus translocation and replication in susceptible cassava and in the resistant lines DSC 167 and DSC 260, to shed light on the tissues and cell types that may be involved in restricting virus movement and replication. To this effect, we used RNAscope^®^ in situ hybridization (ISH), a highly sensitive detection method that was recently developed in our laboratory for virus studies in cassava [14,15] and whose sensitivity can be superior compared to qRT-PCR detection. We further advanced our protocols for more accurate imaging of CBSVs in various cell types using paraffin-embedded tissue sections to study virus invasion of root tissues. We also employed cryo-sectioning to localize the virus in invaded cells which we later want to combine with RNA isolation and RNA-seq to elucidate the molecular basis of the resistance types we have identified.

## 2. Materials and Methods

### 2.1. Cassava Varieties, Viruses and Virus Infection

The African cassava varieties Albert, Namikonga and TMS 96/0304 and the South American cassava germplasm accessions DSC 167 (CIAT, COL 2182) and DSC 260 (CIAT, PER 353) were propagated and kept under glasshouse conditions between 26 °C to 32 °C. The virus isolates, CBSV-Mo83 (DSMZ PV-0949, GenBank accession FN434436) and UCBSV-Ke125 (DSMZ PV-0912, GenBank accession FN433930), were maintained in the cassava variety TMS 96/0304 and further propagated through stem cuttings. Virus infections were established in cassava plants essentially as described in our early study [4], by grafting two buds from CBSVs infected TMS 96/0304 into the stems of experimental plants. After establishment of infections, plants were kept in the greenhouse and symptom development was monitored.

To study virus movement in the accession DSC 167, scions (5 cm, 2 knots) from DSC 167 were side grafted onto healthy TMS 96/0304 rootstocks. Once DSC 167 branches had formed, virus infections were initiated in the chimeric plant by grafting two CBSV-Mo83-infected buds into the lower portion of the stem of the susceptible TMS 96/0304 rootstock. To investigate virus translocation across DSC 167, scions of a healthy TMS 96/0304 were side grafted onto the DSC 167 branch of the chimeric plant TMS 96/0304-DSC 167 (Appendix A).

To study virus movement in DSC 260, scions (5 cm, 2 knots) from TMS 96/0304 were first side grafted onto a healthy DSC 260 rootstock, and allowed to grow and develop into branches in a few weeks. Then, virus infections were initiated by grafting two CBSV-Mo83-infected buds into the branch of the susceptible TMS 96/0304.

### 2.2. Virus Quantification

Total RNA from cassava leaves and stems was extracted using an RNA extraction kit (Epoch, Sugar Land, TX, USA), essentially following the manufacturer’s protocol. RNA purity, quantity and integrity were analyzed in a NanoDrop UV/Vis spectrophotometer (Thermo Fisher Scientific, Waltham, MA, USA). The accumulation of CBSV-Mo83 was measured in a one-step TaqMan qRT-PCR assay (TaqMan kit Maxima Probe/ROX qPCR Master Mix, Thermo Fischer Scientific) with primers, probes and reaction conditions as described before [4]. Cycle threshold (CT) values were used to calculate virus accumulation using the 2^−ΔΔCt^ method [16] with COX (cytochrome oxidase) as a reference gene. Controls (virus negative and positive) were included in every test. Virus expression in the resistant line DSC 167 was calculated relative to the virus titers measured in the infected susceptible TMS 96/0304.

### 2.3. Fixation, Embedding and Sectioning

For virus localization in cassava tissues, the protocol for fixation, embedding and sectioning described by Munganyinka et al. [14] was followed. Stem, leaves and tuberous roots samples were collected from TMS 96/0304-DSC 167 at 7 months after infection (mai) and from DSC 167 and DSC 260 plants 12 mai. Leaf and stem samples of the cassava varieties Albert and Namikonga, either healthy or virus-infected, were taken as controls. Small pieces of plant tissue (5 mm in length) were excised and fixed for 45 min at room temperature (RT) under vacuum and submersion in 10% neutral buffered formalin (Sigma-Aldrich, St. Louis, MO, USA). The incubation step was repeated and after a third replenishment of the formalin solution, the tissue pieces were kept for further 16 h under vacuum. Next, the samples were washed twice in DEPC-treated phosphate-buffered saline buffer (PBS, pH 7.4) for 15 min, followed by dehydration washing in a series of increasing ethanol concentrations (30%, 50%, 70%) for 30 min at each concentration and storage at 4 °C.

For formalin-fixed paraffin embedding (FFPE), 2 steps of infiltration into 95% and 100% ethanol for 30 min each preceded the substitution of ethanol with xylene. This was achieved by stepwise incubation of the explants at RT for 45 min in ethanol/xylene mixtures with decreasing portions of ethanol (2:1, 1:1, 1:2 *v*/*v*) and pure xylene. Xylene was then replaced with paraffin by incubating the samples in mixtures of xylene/paraffin (2:1, 1:1, 1:2 *v*/*v*) and finally in pure paraffin, at 60 °C for 1 h each step. Samples were then transferred into Peel-A-Way molds (Sigma-Aldrich) and kept at RT. Prior to sectioning, paraffin blocks were cooled at 4 °C and sections of approximately 12 µm in thickness were cut using a Microm HM 355 rotary microtome (Thermo Fisher Scientific). Sections were separated from paraffin ribbons, relaxed in a water bath at 37 °C, placed on Superfrost Plus slides (Thermo Fischer Scientific), dried overnight at RT and fixed by baking for 1 h at 60 °C. Directly after baking, sections were deparaffinized by incubating two times in xylene for 5 min each, followed by ethanol washing and storage in the dark.

For cryo-sectioning, ethanol-fixed plant samples were infiltrated sequentially in decreasing ethanol concentrations (50% and 30%) for 30 min each, followed by two steps of incubation in PBS for 15 min at RT. Cryoprotection steps were conducted to prevent structural damages from crystal formation during freezing. To this effect, PBS was complemented with 10%, 20% and 30% sucrose and samples immerged in the solution under vacuum at 4 °C for 3.5 h each step. Embedding was carried out in Optimal Cutting Temperature (OCT) compound (Richard-Allan Scientific™ Neg-50™; Thermo Fisher Scientific). Samples were first transferred to filter paper to absorb excess sucrose from the tissue surfaces, immediately placed into OCT compound and gently mixed. Next, the samples were placed in Peel-A-Way molds complemented with fresh OCT compound and then plunged in liquid nitrogen for several minutes prior to storage at −20 °C. For cryo-sectioning, blocks were placed in a cryostat (CryoStar NX50 equipped with M135 Ultramicrotome blade, Thermo Fisher Scientific) with a specimen temperature set to −20 °C and chamber temperature of −21 °C ± 2 °C. Sections between 15–20 µm in thickness were obtained, immediately placed on Superfrost Plus and Superfrost Plus Gold slides (Thermo Fischer Scientific) and subsequently baked for 1 h at 60 °C to optimize sections attachment.

### 2.4. RNAscope^®^ In Situ Hybridization

The RNAscope^®^ ISH assay was performed as described [14] using the RNAscope^®^ 2.5 HD Detection Reagent-RED kit (cat. no. 322360) from Advanced Cell Diagnostics (Biotechne-ACD, Newark, NJ, USA). Prior to hybridization, slides were baked for 30 min at 60 °C followed by cooling at RT for 20 min, after which a hydrophobic barrier was drawn around the sections using an ImmEdge hydrophobic barrier pen (Biozol Diagnostica Vertrieb GmbH, Eching, Germany); in the case of samples from cryo-sectioning, two additional washing steps with PBS for 5 min each were included directly after the baking step. Next, the sections were treated with hydrogen peroxide provided with the kit for 10 min at RT, to prevent degradation by endogenous peroxidases, followed by washing with distilled water. Directly after, leaf and stem sections were incubated for 15 min in target retrieval buffer warmed up at a temperature in the range of 98–102 °C, while graft-junctions and tuberous roots were incubated in the range of 80–85 °C for 20 min and 5 min, respectively. Tissue sections were dried overnight at RT and treated with the “Protease Plus” solution provided with the kit at 40 °C for 15 min in the case of leaf and stem sections, and for 10 min in the case of tuber sections. Hybridization was performed by incubating the CBSVs-specific probes (ACD cat. no. 509,481 for CBSV-Mo83 and 5,624,391 for UCBSV-Ke125) at 40 °C for 2 h in the hybridization solution provided, followed by washing the sections for 15 min with 2 buffer exchanges. Serial steps of incubation in the provided “AMP” kit reagents were conducted, after which the signal was developed by addition of the Fast-Red substrate, resulting in a red chromogenic signal after processing by the alkaline phosphatase linked to the probes. Slides were subsequently washed in water, counterstained with 50% Gill’s hematoxylin (Sigma-Aldrich) for 2 min and rinsed several times in distilled water. Slides were baked at 60 °C for 45 min, briefly submerged in xylene, covered with EcoMount mounting media (Biocare Medical, Pacheco, CA, USA) and 24 × 40 mm microcover glasses, and air-dried for at least 10 min at RT prior to examination at the microscope.

### 2.5. Histology Staining

Leaves, petioles, stems and tuberous roots sections were used for histological examination and understanding the anatomy of cassava tissues. Cassava sections (12–70 µm in thickness) were stained with FCA (Fuchsin, Crysodin, and Astrablue; Sigma-Aldrich) for 2 min and washed with 100% ethanol. Further staining was in 1% Safranin O (Sigma-Aldrich), and 0.1% Astrablue for 1 min. A third stain was in 0.1 %(*w*/*v*) toluidine blue O (Merck, Darmstadt, Germany) for 1 min and a fourth stain for starch was in Lugol solution (Sigma-Aldrich,) for 5 min. All staining steps were followed by rinsing in water to remove excess stain. After drying at RT, sections were examined under the microscope.

### 2.6. Microscopy Examination

Leaf and stem samples from healthy and infected cassava were examined using a SZX16 stereomicroscope (Olympus, Tokyo, Japan), an Axioscope-A1 (Zeiss, Jena, Germany) and an Axioscope-2 Plus (Zeiss). The “D” setting of the modulator disk was occasionally used to improve the acquisition of the signal.

## 3. Results

### 3.1. The Resistant Cassava DSC 167 Impairs Virus Replication but Not Virus Movement

The infection process of CBSV-Mo83 in the chimera plant TMS 96/0304-DSC 167 was followed. Symptoms on leaves were observed 14 days post inoculation (dpi) in the susceptible rootstock TMS 96/0304 followed by the occurrence of brown streaks visible on the stem close to the graft intersection between TMS 96/0304 and the DSC 167 at 20 dpi. In contrast, leaves and stem of the resistant DSC 167 remained symptomless during the entire 5-6 months of observations in accordance with our previous study [4]. CBSV translocation was followed in TMS 96/0304-DSC 167 chimeric plants by dissecting the branch that had developed from the DSC 167 scion, in sequential steps of 1 cm, starting from the graft junction up to the tip of the branch and subjecting the Total RNA to TaqMan qRT-PCR (Figure 1).

At the graft intersection area, severe necrosis of the TMS 96/0304 tissues was noticed (Figure 1A) and a high virus load was found (Figure 1B, red bar and sample 1). In distant regions from the graft intersection, the virus titer instantly decreased to almost undetectable values around the threshold of the qRT-PCR and was nil in many sections from DSC 167 branch (Figure 1B, open circles). The erratic detection of the virus in stem slices suggested that the virus was translocating through the stem but not accumulating in cells and tissues.

A next bioassay step gave further proof that long-distance virus movement in the resistant DSC 167 was not impaired (Figure 2). In these experiments, a scion of a healthy TMS 96/0304 was side grafted onto the symptomless DSC 167 branch (Figure 2C) of the chimeric TMS 96/0304-DSC 167 plant, and approximately 20 days after grafting (dag) leaves developing on the growing susceptible scion started to show distinct CBSV symptoms (Figure 2D).

Cross sections prepared from branches of CBSV-infected TMS 96/0304 and DSC 167 were subjected to RNAscope^®^ ISH using a probe against the CBSV-Mo83 P1 gene. Examination of TMS 96/0304 sections revealed strong signals distributed throughout the tissues as red dots or clusters of dots in external and internal phloem cells (Figure 3B, white arrows) and in the parenchyma (Figure 3B, red arrow). There was no background signal in stem sections of the non-infected TMS 96/0304 (Figure 3A), showing the high specificity of the detection method. When sections of DSC 167 were examined, a red signal indicating virus RNA accumulation was observed, in a minimal amount and limited to the external phloem only (Figure 3C–F, white arrows). There was no trace of signal in the internal phloem and the parenchyma cells (Figure 3C–F).

An examination of virus localization at the graft intersections between TMS 96/0304 and DSC 167 showed that CBSV-Mo83 had fully invaded TMS 96/0304 tissues (Figure 4A,C), while in the DSC 167 tissues the virus remained localized in the external phloem only (Figure 4B,C). There were considerable necrotized tissues in TMS 96/0304, as shown by the evident brownish necrotic areas visible by light-microscopy at the graft intersections (Figure 4D, arrows).

### 3.2. The Resistant Cassava DSC 167 Restrains CBSV Replication and Confines the Virus to the Phloem

When DSC 167 leaves were examined to trace the virus, it was necessary to inspect sections taken from several positions along the branch because of the very low virus content. In contrast, cross sections of leaves taken from the cassava variety Albert showed bright red ISH signals dispersed broadly over the entire leaf midvein section (Figure 5C). Viral RNA in DSC 167 was only rarely found in cross sections of DSC 167 leaf midribs and associated with phloem cells of the abaxial phloem and occasionally also in cortical cells, with limited distribution (Figure 5B).

RNAscope^®^ ISH of cross sections across the entire leaf blades further corroborated that there were only traces of CBSV in DSC 167. The cassava varieties Albert and Namikonga taken as controls showed red signals evenly dispersed over the sections. In panorama pictures stitched from individual sections, the presence of red clusters in phloem cells and in palisade, mesophyll and midrib tissues showed that the virus had established itself in the susceptible variety Albert (Figure 6A). Similarly, in the resistant variety Namikonga, there was no association of hybridization signals with particular cells; however, the CBSV invasion appeared patchy and there was a strong signal in some areas of the leaf blade while others were entirely free (Figure 6B). The low virus titers measured by qRT-PCR in Namikonga [4] would then indicate a patchy plant invasion (movement) rather than providing evidence that virus replication is generally reduced in this host. In cross sections of DSC 167 leaves, virus-specific signals were generally absent and could only be visualized at high resolution, appearing as very few scattered dots localized in midrib sections (Figure 6C, white arrows.)

Examination of cross sections from root tubers of CBSV-Mo83 infected DSC 167 further confirmed that also tubers had no symptoms and were free of any necrotic tissue (Figure 7A). RNAscope^®^ ISH hybridization signals, albeit at very low frequency, were nevertheless found and distinct red dots traced the virus only into the phloem cells (Figure 7B–D). No signal was observed in sections of uninfected tuberous root (Figure 7E). In contrast, in tuber sections from TMS 96/0304, CBSV hybridization signals were broadly distributed and the virus accumulated in phloem and non-phloem cells (Figure 7F, red arrows).

### 3.3. The Resistant Cassava DSC 260 Restricts Virus to the Phloem and Remains Free of Disease Symptoms

Upon CBSV infection by bud grafting, DSC 260 plants remained symptomless and there were no symptoms visible on all leaves, stems and tubers. In leaf sections subjected to RNAscope^®^ ISH, the upper and the middle leaves were generally free of any signal. However, in the lower leaves, close to CBSV-infected buds, red signals were occasionally found in abaxial and adaxial phloem cells (Figure 8A, white arrows) and in cells of the spongy parenchyma (Figure 8A, red arrow). In contrast, viral RNA was present in stem sections as single red dots or small clusters located in the external and internal phloem (Figure 8B,C, white arrows) and in the phellogen (Figure 8B, yellow arrows). In the susceptible TMS 96/0304, the virus was found in all cell types at a high intensity (Figure 8 D).

There were no necrosis symptoms visible in any of the tuberous roots of the three DSC 260 plants examined. However, viral RNA was found in abundance in tuberous tissue (Figure 9A,B, white arrows), where red dots indicating virus presence were localized in phloem cells only and no invasion of parenchyma cells was observed.

To verify virus movement in DSC 260, a TMS 96/0304 scion was side grafted onto the infected DSC 260 rootstock and the developing buds from this branch were subjected to RNAscope^®^ ISH. Extremely high CBSV-Mo83 signals were found in all cell types of the developing buds (Figure 10B,C). This was in sharp contrast to the tissue section prepared from the DSC 260 branch (Figure 10A), where the virus was restricted to phloem tissue only.

Thus, experimental evidence was provided to show that virus movement was not compromised in DSC 260 and despite the high virus pressure from virus replication in a branch of the permissive TMS 96/0304, there was no evidence for virus replication and accumulation in DSC 260 leaf tissues. CBSV accumulated but remained localized in phloem tissues of cassava stem and roots; however, there was no sign of disease symptoms in either of the organs.

### 3.4. The Resistant Cassava Line DSC 167 Allows UCBSV Infection but Contains Virus around Necrotic Lesions

The line DSC 167 was previously identified as a highly resistant cassava line that neither became infected with CBSV nor with UCBSV [4]. When grafted with UCBSV-Ke125, this cassava line developed necrotic halos around the inserted buds, a symptom that was never observed with graft-infections of CBSV-Mo83. Under particular experimental conditions, when less than 2-month-old cassava plantlets were grafted with UCBSV-Ke125 scions, DSC 167 plants became infected, albeit at very low efficiency (3/10 plants). In the case of an infection, a specific response to the virus was seen on the leaves, starting with small chlorotic spots (Figure 11A) that became enlarged over time to turn into necrotic spots with necrotic circles forming halos (Figure 11B–E) and eventually developing into chlorotic/necrotic patches. The symptoms associated with this artificial infection were not reproduced with any CBSV isolate and were also very different from the typical leaf symptoms of UCBSV-Ke125 in a susceptible variety (Figure 11F).

We studied this phenomenon by subjecting leaf, stem and root tissues to RNAscope^®^ ISH using a probe against the UCBSV-Ke125 P1 gene. Prior to the assay, sections from the same tissue were prepared either by paraffin embedding or by cryo-sectioning to compare both methods. In DSC 167 leaves, UCBSV was mostly located in necrotic areas indicated by broadly distributed red signal clusters across all cell and tissue types in FFPE sections (Figure 12Ap,Bp) and cryo-sections (Figure 12Ac,Bc). In necrotic spot tissues, red signals were observed as small restricted clusters in both FFPE sections (Figure 12Cp) and cryo-sections (Figure 12Cc). Additionally, signals were observed in non-symptomatic leaf areas as small red dots, representing single viral RNA molecules; this pattern however was only observed in FFPE leaf sections (Figure 12Dp, white arrows) but not in RNAscope^®^ ISH of respective cryo-sections (Figure 12Dc).

In stem sections from this UCBSV-Ke125-infected DSC 167, red signals were observed in the cortex, in the internal and external phloem and in the parenchyma cells both in FFPE sections (Figure 13Ap–Cp) and in cryo-sections (Figure 13Ac–Cc).

Satisfactory preparations of sections from tuberous roots, in particular from those with abundant necrotic tissue, were only achieved when root tissue was embedded in paraffin. Cryo-sections from those tissues readily tore apart because of the large areas composed of soft parenchyma cells and disintegrated necrotized tissue structures, impeding the obtainment of samples for further processing.

Roots of DSC 167 experimentally infected with UCBSV-Ke125 were severely affected by the disease and large necrotized areas across the tuber were observed (Figure 14A). The virus hybridization signal was widely distributed as red clusters in phloem and non-phloem cells, with very high signal intensities in necrotic areas (Figure 14B–D).

Subjecting the otherwise resistant cassava DSC 167 to artificial infection conditions to establish an infection with UCBSV revealed a tissue invasion that was very different from that of a susceptible cassava line. The initial chlorotic spot symptoms visible on leaves developed further into necrotic spots. The virus remained localized in the areas surrounding the necrotic spots, indicating that the plant responds to virus infection by confining the virus. This specific case presented is in contrast to the general observation that in DSC 167 U/CBSV do not replicate and virus infections cannot be established.

## 4. Discussion

The resistance identified in cassava germplasm from South America against cassava brown streak viruses [4] was further studied in DSC 167, a line that CBSV did not infect and in DSC 260 in which a CBSV infection remained restricted to root tissues with no symptoms visible on leaves. CBSV infections were followed in the two cassava lines by providing a constant virus influx from a susceptible source. Otherwise, virus infections would not be maintained in these lines and even under these conditions, virus resistance persisted. CBSV, like other plant viruses, is transported through the plant predominantly with the source-sink flow of photo assimilates [17,18,19,20]. For systemic infection and long-distance movement, CBSV needs to reach the phloem and has to pass from sieve elements into the companion cells to translocate to parenchyma and mesophyll cells for replication. Our molecular tests and bioassays provided evidence that CBSV-Mo83 can translocate through the vascular tissue (long-distance movement) of the resistant cassava DSC 167. However, it did not cause symptoms and there was no virus replication. Phloem movement of the polerovirus potato leafroll virus (PLRV) was also found when a resistant potato variety “Bismarck” was grafted in between a healthy rootstock and a virus-infected scion. Similarly, the virus was translocated through the phloem and infected the rootstock but did not replicate in the resistant variety [21].

Our bioassays provided evidence for translocation of viable virus; however, it was very challenging to trace the virus in the phloem by qRT-PCR [4]. RNAscope^®^ ISH provided very sensitive detection of even minute amounts of virus and confirmed that the virus was indeed present in the South American cassava line DSC 167. In stem tissue sections of a DSC 167 branch side grafted on a CBSV-infected TMS 96/0304 rootstock, viable virus was detected but there was no accumulation of virus in phloem cells and furthermore, virus invasion and replication in parenchyma tissues was not evident. According to these results, it appears that phloem unloading of CBSV is prohibited in this line. In contrast to the phloem-restricted virus PLRV, the ipomoviruses U/CBSV replicate in mesophyll cells and exit the vascular tissues to invade and replicate in non-vascular tissues. Thus, phloem restriction is not a constitutive feature associated with the viruses but rather a specific resistance response of the host since the viruses otherwise move and replicate to a high extent in susceptible cassava lines.

Histological examination of sections prepared from different organs revealed the presence of two types of phloem in cassava sections (Appendix A): the external phloem with an annular arrangement and the internal phloem with a cord-like internal phloem adjacent to the pith [22,23]. The internal phloem can serve as an enhancer for conduction and as a storage organ [24]. Internal and external phloem are implicated in long-distance virus movement in two directions following the flow of assimilates. Upward virus movement occurs in the internal phloem and is faster compared to the downward movement. This was also described for pepper mottle virus (PepMoV) and tobacco mosaic virus (TMV) [17,20] and, thus, specific routes exist by which viruses move in the two phloem types that are otherwise barely connected and with only little cross traffic [20]. The study with PepMoV suggests connections in roots and in nodes of the cotyledon and nodal regions of the stems and the hypocotyl of tomato, potato and other Solanaceae [17]. In cassava, we assume that external and internal phloem converges in buds where the inner phloem is not yet differentiated.

In susceptible cassava lines, CBSV was found in all cells and tissues and in both phloem types as well as in cortical cells (phellogen). In contrast, CBSV was only detected in the external phloem of the resistant DSC 167. This was also shown for cucumber mosaic virus (CMV), potato leaf roll virus (PLRV) and pepper mottle virus (PepMoV) using both external and internal phloem for directed movement [25,26]. In a resistant Capsicum sp., CMV was detected in the external phloem only [26]. Pepper mottle virus Florida isolate (PepMoV-FL) was able to move downwards through the external phloem in the resistant Capsicum annuum cv. Avelar, while upward movement through the internal phloem was restricted, resulting in virus-free newly sprouting tissues [26]. In contrast, PLRV remained confined to the internal phloem of a resistant potato [27]. Interestingly, melon necrotic spot virus (MNSV) used the external phloem during early stages of invasion to switch to the internal phloem during later stages of infection [28]. Taken together, virus resistance in DSC 167 occurs with a block of the upwards movement of CBSV in the internal phloem.

Resistance to long-distance movement is achieved either by preventing the virus from entering the sieve element-companion cells (SE-CC) (loading) or by preventing the virus from exiting the SE-CC complex to enter into sink tissues (unloading) [29,30]. In our study, U/CBSV was directly introduced by bud grafting into the phloem of cassava, thus it was impossible to prove that virus loading into the phloem was restricted. However, RNAscope^®^ ISH clearly demonstrated that in tissue sections of both DSC 167 and DSC 260 the virus was effectively contained in the phloem. In the resistant cassava DSC 260, despite being absent in leaves, CBSV was able to translocate from the roots through the stem of DSC 260 to infect a TMS 96/0304 scion and this long-distance movement was in both directions, in the external and internal phloem. Furthermore, compared to only traces of virus visible by RNAscope^®^ ISH in DSC 167, there was a considerable amount of virus in the phloem of DSC 260 stem and root tuber sections. Thus, from the density of hybridization signals, it can be assumed that virus replication in DSC 260 phloem companion cells happens. Consequently, inhibition of phloem unloading is a likely explanation for the resistance observed in the immune DSC 167 and the resistant DSC 260 alike.

Resistance against plant diseases is the result of an interaction between host (genotype) and pathogen (strain, pathovar) modulated by developmental stages of the host and the environment [31]. The cassava line DSC 167 did not become infected with UCBSV and, like CBSV, virus was found only in phloem cells, provided the infection source was present. In contrast, under experimental conditions, very young cassava plants grown at temperatures >30 °C became infected with UCBSV and the virus was able to exit the phloem and replicate in the parenchyma cells. This phenomenon was very rare and only seen under laboratory conditions but it showed that this phenomenon is modulated by plant age and environment. Age-related resistance (ARR) is found in many pathosystems including bacteria, fungi, oomycete, insects and viruses. In *Arabidopsis* sp., *Pseudomonas syringae* caused infections in young plants by suppressing the salicylic acid SA pathway. Upon maturation, bacterial infections declined rapidly as a result of age-related resistance responses [32]. Similarly, tomato plants infected with TMV were highly susceptible when infected within the first few weeks after germination while 6 weeks after germination TMV inoculations were not successful [33,34]. However, high susceptibility in early developmental stages can only partly explain the response of DSC 167 against UCBSV since this phenomenon could not be reproduced by infecting with CBSV.

Hypersensitivity (HR) indicates an incompatible plant/virus interaction. After virus recognition and primary replication, localized necrotic lesions develop in inoculated leaves to restrict the virus to the point of entry and prevent virus movement and invasion to neighboring cells. In systemic hypersensitive reactions (SHR), the signal spreads along with the virus and more cells eventually become necrotized [10,35,36]. The UCBSV resistance response in DSC 167 was associated with callus formation around the buds inserted that eventually necrotized. This strong hypersensitive reaction may restrict virus entry and further spread throughout the plant. In contrast, there was no necrosis at the bud insertion site when very young cassava plants were infected with UCBSV and the virus moved through the phloem and unloaded into parenchyma cells to replicate. The infected plants responded with chlorotic spots turning into necrotic circles that restricted virus movement to neighboring cells. We speculate that similar to other viruses [37,38,39], this unique response in DSC 167 to UCBSV infection may also be due to temperature-modulated effects; however, more experiments are needed to provide evidence for these observations.

In this study, we provide an explanation for the resistance identified in some South American cassava germplasm lines against the viruses causing CBSD. While this is a decisive step towards the development of U/CBSV resistant varieties, there are still many open questions around our observation. In both cassava lines, the virus infections are restricted to the phloem cells, but only in DSC 260 does virus accumulation indicate virus replication. Are there two discrete mechanisms acting, inhibition of replication and phloem restriction? What is the specific function of the internal phloem in this virus resistance? Structural differences and functions for these phloem types were shown in pumpkin. The proteins PP1 and PP2 were found in both phloem types but only the extrafascicular phloem released mature proteins, despite both mRNAs occurring in both phloem [40,41]. It will be interesting to see if similar phloem proteins regulate phloem transport/translocation in resistant cassava. Our cryo-sectioning protocol for cassava tissues may support transcriptome and proteome profiling of dissected phloem cells for resistance studies. A focus on cassava’s tuberous roots may then shed light on the critical role of the tuberous roots in the cassava brown streak virus disease.

## Figures and Tables

**Figure 1 cells-10-01221-f001:**
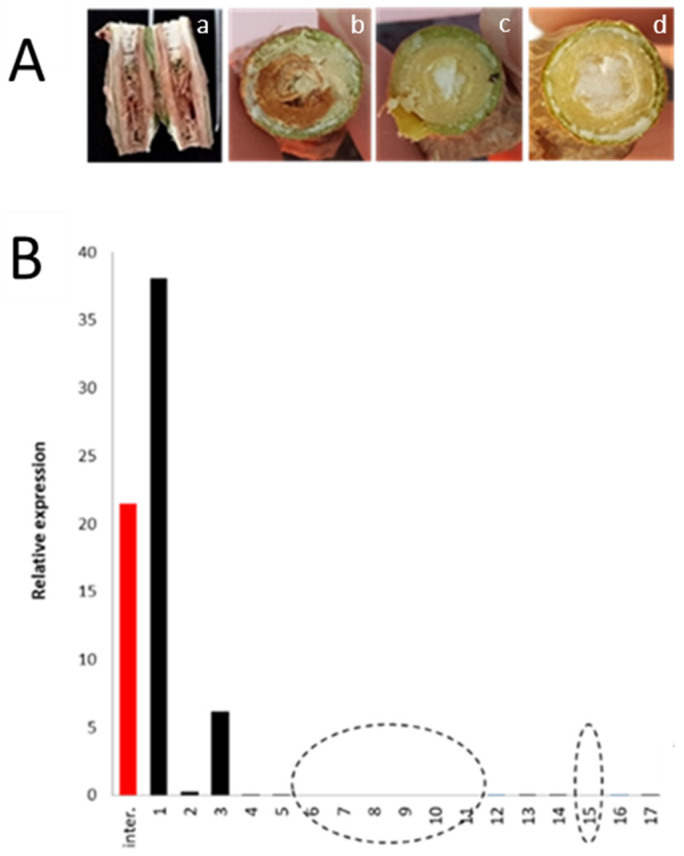
CBSV symptoms and virus quantification in stem sections of a DSC 167 branch from the chimeric TMS 96/0304-DSC 167. (**A**) Necrosis in DSC 167 was only found at the graft intersections where virus was readily detected (**a**,**b**), but not in regions of the stems distant from the graft junction (**c**,**d**). (**B**) Virus detection by TaqMan qRT-PCR in single slices of sections taken from a DSC167 branch starting at the graft intersection up to the top, at 1 cm intervals. Open circles mark samples (branch areas) where the assay was negative. Overall, DSC 167 branches of 4 plants were individually tested, with negative-assay results dispersed over the length of the branch.

**Figure 2 cells-10-01221-f002:**
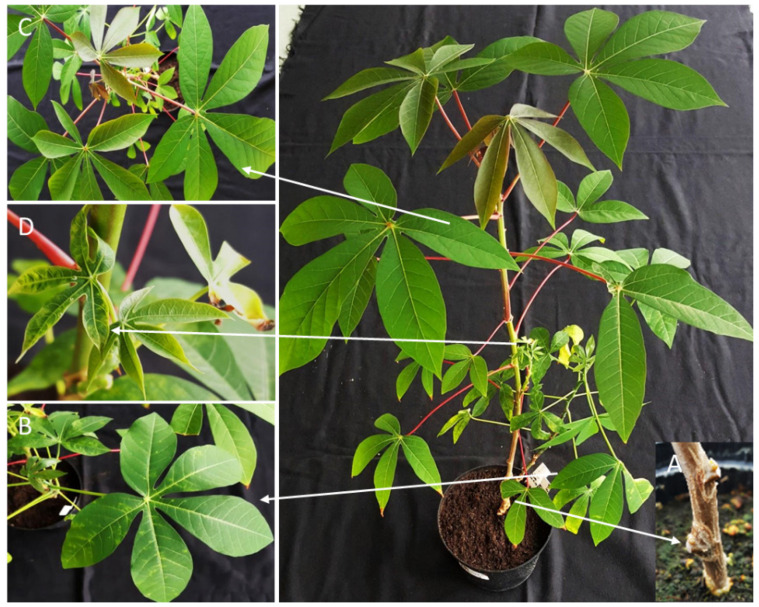
Virus movement through a resistant cassava DSC 167. A TMS 96/0304-DSC 167 chimeric plant was infected with CBSV-Mo83 by inserting 2 buds onto the TMS 96/0304 rootstock (**A**); typical CBSV symptoms developed on the sensitive TMS 96/0304 12 dag (**B**). Leaves developing on the branch of the resistant DSC 167 remained symptomless (**C**), while leaves of the TMS 96/0304 scion grafted into the DSC 167 branch showed distinct CBSV symptoms, proving virus translocation through DSC 167 (**D**).

**Figure 3 cells-10-01221-f003:**
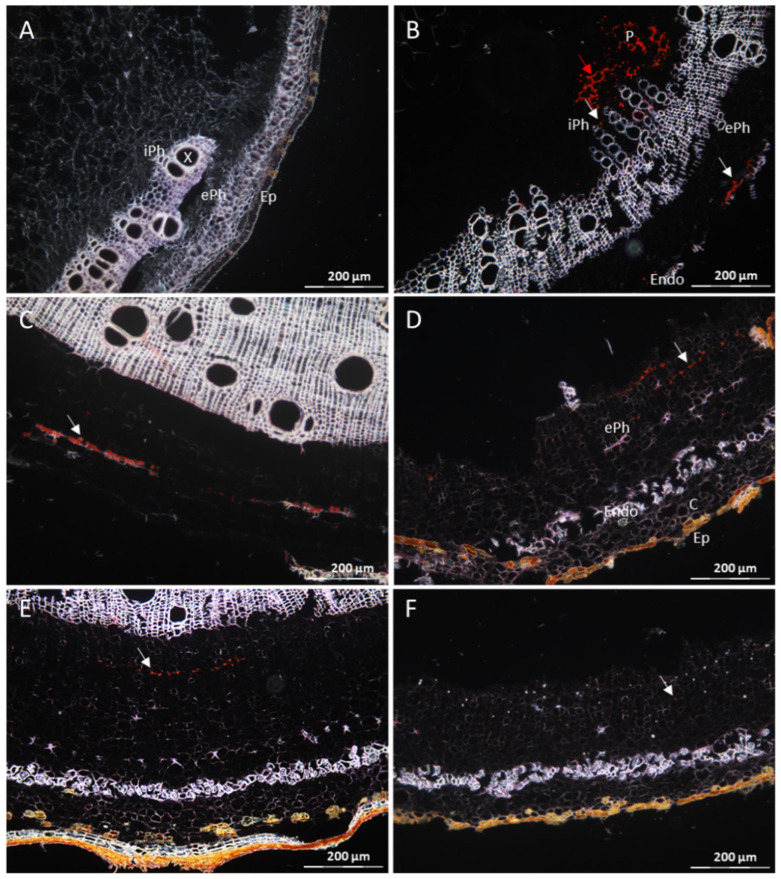
Distribution of CBSV-Mo83 RNA in cross-tissue sections of DSC 167 and TMS 96/0304 cassava stems. Cross sections of cassava stem tissues subjected to RNAscope^®^ ISH using a CBSV-Mo83 probe. (**A**) Absence of signals in stem sections of uninfected TMS 96/0304. (**B**) Red dots indicate presence of viral RNA in phloem cells (external and internal phloem, white arrows) and in parenchyma cells of TMS 96/0304 (red arrow). (**C**–**F**) In DSC 167, virus signals are found in external phloem only ((**C**–**F**), white arrows). X, xylem; P, parenchyma; C, cortex; ePh, external phloem; iPh, inner phloem; Ep, epidermis; Endo, endodermis.

**Figure 4 cells-10-01221-f004:**
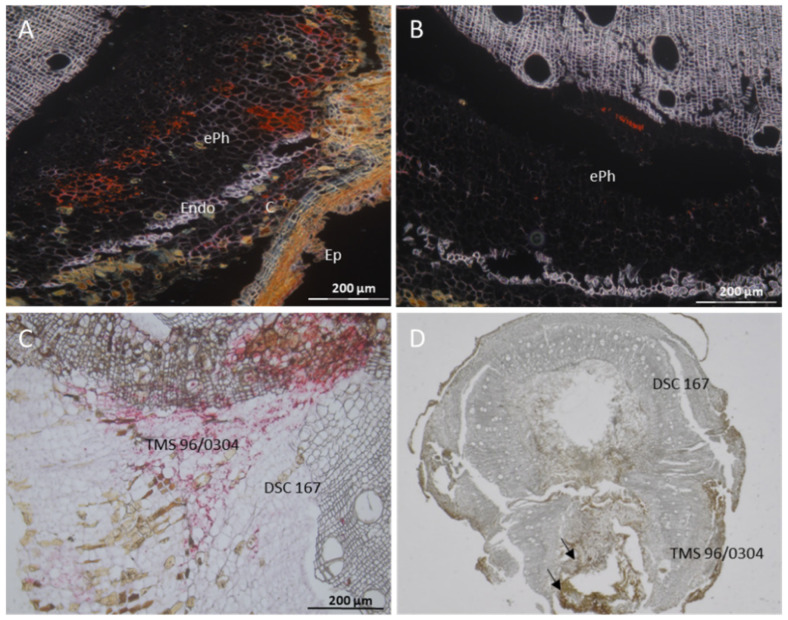
Distribution of CBSV-Mo83 RNA in cassava tissue at a graft intersection of the chimeric plant TMS 96/0304-DSC 167. Cross sections of cassava stem tissue subjected to RNAscope^®^ ISH using a CBSV-Mo83 probe show red signals indicating the presence of the viral RNA. (**A**) Virus presence in phloem and non-phloem cells of TMS 96/0304. (**B**) Virus found only in the external phloem of DSC 167. (**C**) Cross section through the graft intersection with red signals scattered throughout TMS 96/0304 portion and rarely found in DSC 167. (**D**) Anatomy of the cassava graft intersection in the chimeric TMS 96/0304-DSC 167. Arrows point to necrotized tissues. C, cortex; ePh, external phloem; iPh, inner phloem; Ep, epidermis; Endo, endodermis.

**Figure 5 cells-10-01221-f005:**
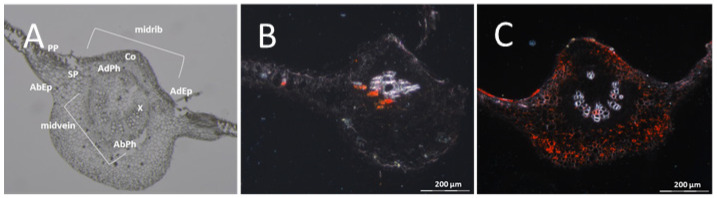
Distribution of CBSV-Mo83 RNA in cross sections of a cassava leaf from susceptible (Albert) and resistant lines. (**A**) Overview on the anatomy of a cassava leaf midrib. (**B**) Presence of the viral RNA mostly in the abaxial phloem cells in DSC 167 midvein. (**C**) Presence of viral RNA in phloem and non-phloem cells in Albert midvein. X, xylem; PP, palisade parenchyma; SP, spongy parenchyma; AdEp, adaxial epidermis; AbEp, abaxial epidermis; AbPh, abaxial phloem; AdPh, adaxial phloem; Co, collenchyma cells.

**Figure 6 cells-10-01221-f006:**
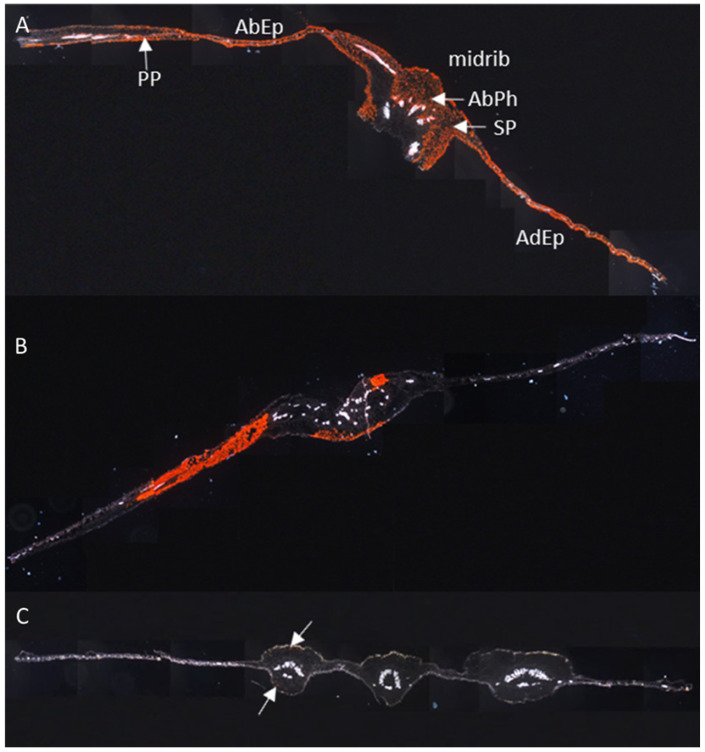
Distribution of CBSV-Mo83 RNA in cassava leaf sections. Panorama pictures were composed of cross sections from leaf tissues subjected to RNAscope^®^ ISH using a CBSV-Mo83 probe. Red dot signals indicate the presence of viral RNA in phloem cells and non-phloem cells of the cassava variety (**A**) Albert, (**B**) Namikonga and (**C**) DSC 167, the latter showing signal only in few cortical collenchyma cells (white arrows). PP, palisade parenchyma; SP, spongy parenchyma; AdEp, adaxial epidermis; AbEp, abaxial epidermis; AbPh, abaxial phloem; AdPh, adaxial phloem.

**Figure 7 cells-10-01221-f007:**
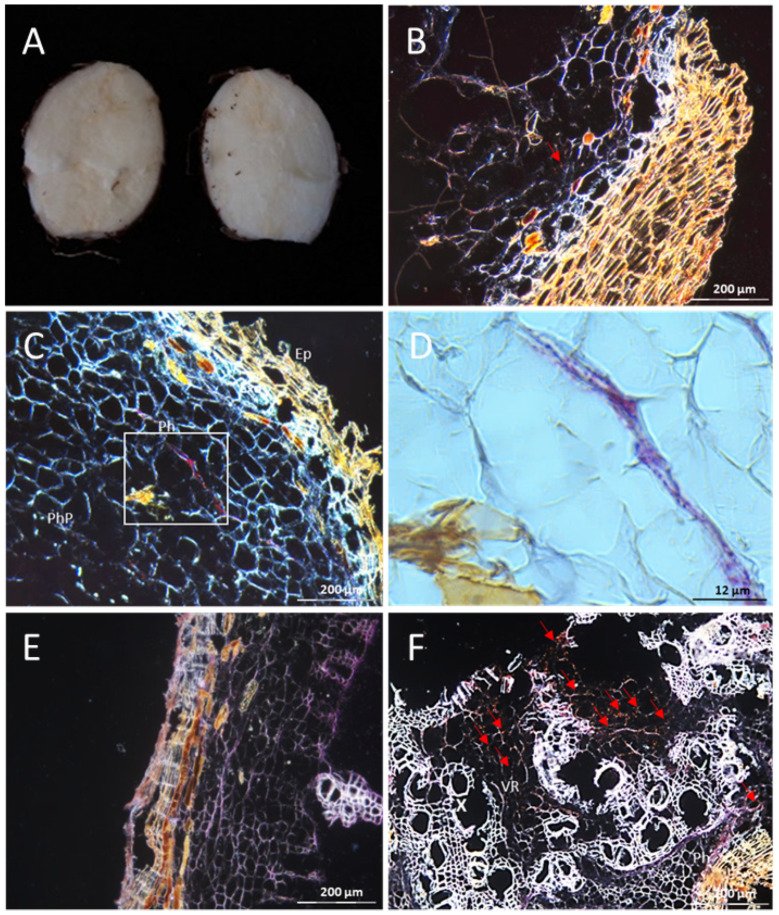
Distribution of CBSV-Mo83 RNA in cross sections of DSC 167 and TMS 96/0304 tuberous roots. (**A**) Longitudinal cutting in DSC 167 cassava tuberous roots. (**B**,**C**) Cross sections of cassava DSC 167 root tuber subjected to RNAscope^®^ ISH using a CBSV-Mo83 probe showing few red dots in phloem cells (arrow). (**D**) Close–up images of (**C**). (**E**) Section of an uninfected tuberous root subjected to RNAscope^®^ ISH. (**F**) Virus presence in phloem and non-phloem cells (arrows) of TMS 96/0304. X, xylem; VR, vascular ray parenchyma cells; Ph, phloem; Ep, epidermis; Exo, exodermis; PhP, phloem parenchyma.

**Figure 8 cells-10-01221-f008:**
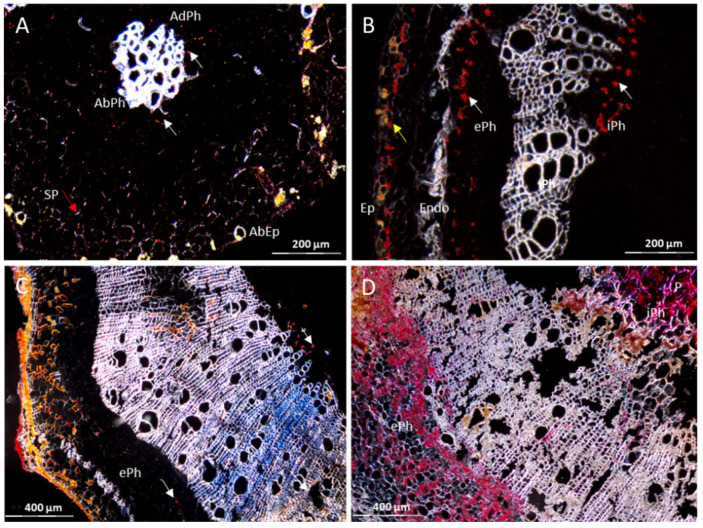
Distribution of CBSV-Mo83 RNA in cross sections of cassava DSC 260 and TMS 96/0304. Red dots and clusters indicate the presence of viral RNA in (**A**) leaf sections and (**B**,**C**) stem sections of DSC 260 cassava; specifically, in phloem cells (white arrows), non-phloem cells (red arrow) and the phellogen (yellow arrow). (**D**) Stem section of TMS 96/0304 showing intense and widely dispersed virus signals. SP, spongy parenchyma; AbEp, abaxial epidermis; AbPh, abaxial phloem; AdPh, adaxial phloem; P, parenchyma; ePh, external phloem; iPh, inner phloem; Ep, epidermis; Endo, Endodermis.

**Figure 9 cells-10-01221-f009:**
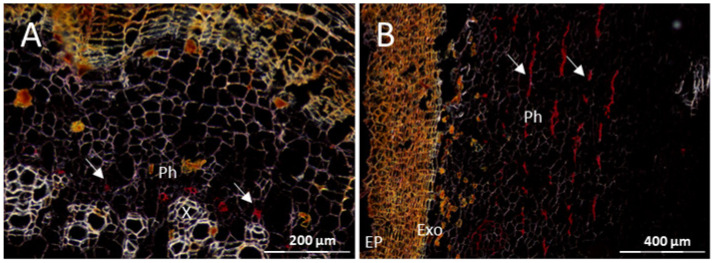
Distribution of CBSV-Mo83 RNA in cross sections of cassava DSC 260 tuberous roots. Red dots indicate viral RNA in phloem cells only ((**A**,**B**); white arrows). X, xylem; Ph, phloem; Ep, epidermis; Exo, exodermis.

**Figure 10 cells-10-01221-f010:**
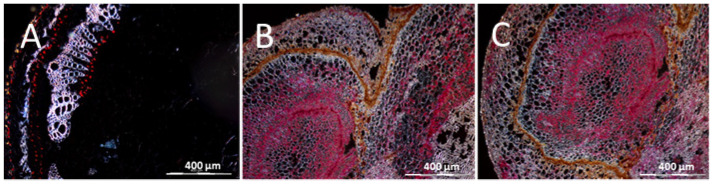
Distribution of CBSV-Mo83 RNA in cross sections from DSC 260 and TMS 96/0304 buds. (**A**) Red signals in DSC 260 bud tissues, restricted to phloem and cortical cells only. (**B**,**C**) Strong signals dispersed over all cell types indicate highly abundant viral RNA in buds of the susceptible cassava TMS 96/0304.

**Figure 11 cells-10-01221-f011:**
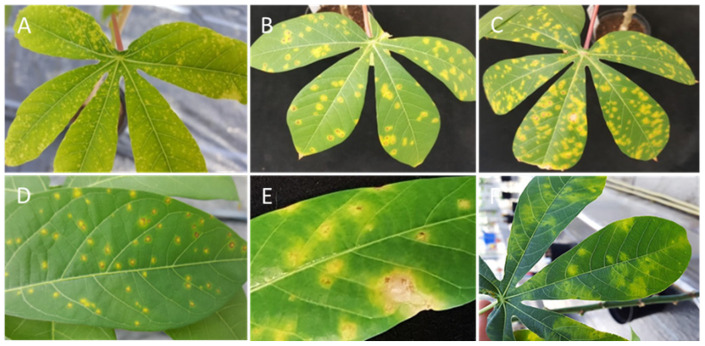
Differential response of cassava DSC 167 and TMS 96/0304 to infections with UCBSV-Ke125. The infection in DSC 167 starts with (**A**) yellow dots in the leaf that develop to (**B**,**C**) more significant yellow spots surrounded by necrotic spots. (**D**,**E**) Close-up of necrotic spots caused by UCBSV-Ke125. (**F**) Typical UCBSV-Ke125 symptoms on a susceptible TMS 96/0304.

**Figure 12 cells-10-01221-f012:**
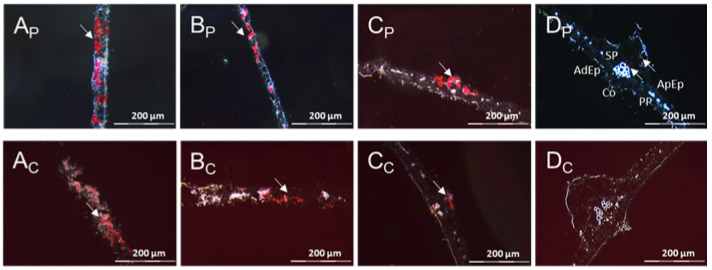
Distribution of UCBSV-Ke125 RNA in cross sections of DSC 167 cassava leaf tissues. Cross sections from cassava subjected to RNAscope^®^ ISH using an UCBSV-Ken125 probe. Red signal indicates the presence of the viral RNA in phloem cells and non-phloem cells (white arrows). (**Ap**–**Dp**) Sections prepared from samples embedded in paraffin. (**Ac**–**Dc**) Tissues subjected to cryo-sectioning. X, xylem; PP, palisade parenchyma; SP, spongy parenchyma; AdEp, adaxial epidermis; AbEp, abaxial epidermis; AbPh, abaxial phloem; AdPh, adaxial phloem; Co, collenchyma cells.

**Figure 13 cells-10-01221-f013:**
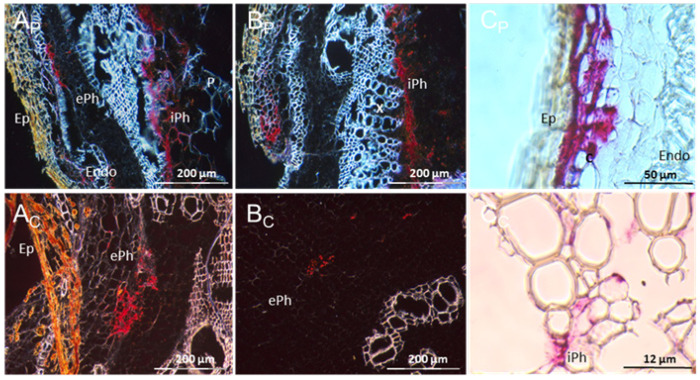
Distribution of UCBSV-Ke125 RNA in cross sections of DSC 167 stem tissues. Cross sections were subjected to RNAscope^®^ ISH using an UCBSV-Ke125 probe, showing red signals indicating the presence of the viral RNA in phloem cells and non-phloem cells. (**Ap**–**Cp**) Sections prepared from samples embedded in paraffin. (**Ac**–**Cc**) Tissues subjected to cryo-sectioning. X, xylem; P, parenchyma; C, cortex; ePh, external phloem; iPh, inner phloem; Ep, epidermis; Endo, endodermis.

**Figure 14 cells-10-01221-f014:**
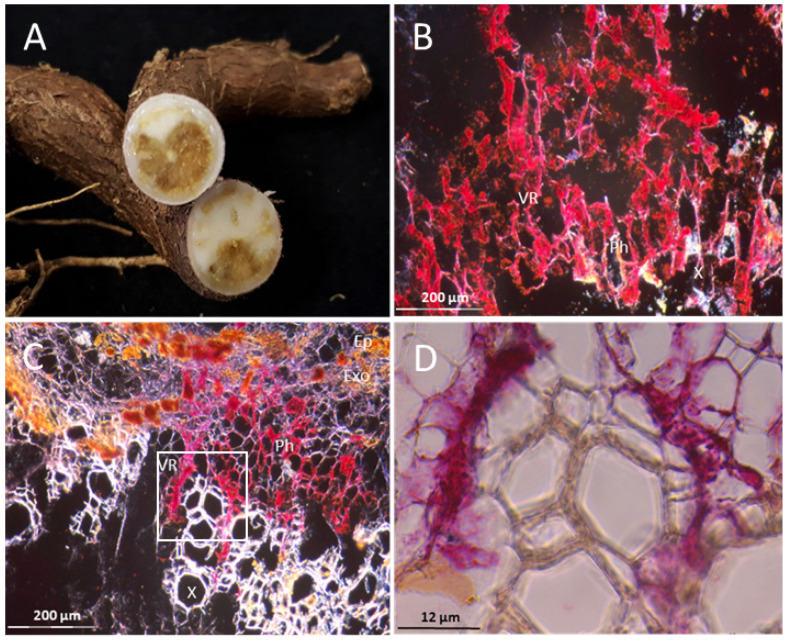
Distribution of UCBSV-Ke125 RNA in cross sections of DSC 167 root tubers. (**A**) Cutting across DSC 167 tubers, showing severe necrosis symptoms. (**B**,**C**) RNAscope^®^ ISH signals in red clusters distributed in all cell types except the xylem. (**D**) Close–up images of (**C**). X, xylem; VR, vascular ray parenchyma cells; Ph, phloem; Ep, epidermis; Exo, exodermis.

## Data Availability

The data (raw figures) presented in this study are available on request from the corresponding author.

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
