# Peer review of "Differential Tropism in Roots and Shoots of Resistant and Susceptible Cassava (Manihot esculenta Crantz) Infected by Cassava Brown Streak Viruses"

_cells, 2021, doi:10.3390/cells10051221_

Round 1

Reviewer 1 Report

Cassava brown streak virus (CBSV) is a positive ssRNA ipomovirus that renters the cassava tubers not fit for human consumption. Genetic resistance to CBSV is scarce and is highly desired.

The present manuscript is based on the findings described in a paper from the same group, published two years ago in "Frontiers in Plant Science": Resistance against cassava brown streak viruses from Africa in cassava germplasm from South America. In this paper the authors found:

Since African germplasm is susceptible to CBSV, the authors searched for CBSV resistance in South American germplasm: upon graft infection. seven cassava lines, including lines DSC167 and DSC118, had no symptoms in tuberous roots when infected with a number of virus isolates. In these plants, virus was undetectable in all tissues tested by RT-PCR. By grafting the susceptible infected cv Albert on line DCS167, the authors found that virus replication is inhibited in line DCS167, but not virus movement.

Here the authors aimed at finding the mechanism leading to immunity to CBSV infection. The rationale was that the virus was confined to tissues where it could not replicate, move out of the infected cells and spread long-distance. Using various combinations of grafts, they used FISH to detect the virus in the tissues of the various cassava lines - the immune DSC167, the line with virus restricted to roots DSC260 - to find out where is the block preventing the virus to replicate and spread. They found: 1) the resistant line DSC167 was devoided of virus, except at the graft intersection with infected susceptible germplasm (necrotic tissues), indicating an absence of long-distance transport of the virus. CBSV was not detected in the internal phloem and in the parenchymal cells. 2) DCS167 does not support virus replication; the virus was detected only in the external phloem in contrast to susceptible germplasm were the virus was detected in abundance in the two phloem types. A heavy load of viral RNA was detected in tubers of susceptible germplasm.

Altogether, the authors explain the resistance previously described in line DCS167. It is expected that this line will help alleviate cassava disease in East Africa. This makes this manuscript recommendable for publication as is.

Author Response

Thank you very much for your kind attention and great efforts to review our manuscript. We are very grateful for your positive response supporting a publication that can have a critical impact on future cassava breeding and improvement

Reviewer 2 Report

The authors show convincingly by RT-PCR and grafting experiments (Fig. 1-2) that there is hardly any virus in DSC167 stems and that the virus detected in DSC167 stems is probably only on passage. Next they show by RNAScope enzymatic labeling of stem sections that there is more virus RNA in stems of a susceptible cultivar, where label is found in phloem and parenchyma. In DSC167 stems, only label of external phloem and occasionally cortical cells is observed. The “resistant” variety Namikonga showed patchy distribution of virus in leaves, susceptible cultivars show label in all tissues.

The RNA label in DSC260 is stronger than in DSC167 and also epidermal and parenchyma cells are infected. Grafting experiments show that virus phloem movement is not impeded in DSC260.

Finally, the authors graft-inoculate DSC167 with UCBSV. If I understand well, infection only occurred when grafting young plants. When infected, leaves show necrotic spots. RNAScope shows infection not only of phloem, but also of other tissues in leaves and stems. Roots seem to be severely infected by UCBSV.

The authors discuss that resistance is probably due to phloem restriction, more precisely inhibition of phloem unloading, of the otherwise all tissues infecting virus.

You claim in the abstract that no CBSV resistance is known but then you present an immune variety (DSC167). This is somewhat contradictory. It seems that this is true for African cassava but not for South American varieties. Please clarify.

Please add readable scale bars to all figures showing microscopic images.

Line 159: What is the peroxide concentration?

Line 264: Title 3.2 At best, you present indirect evidence that there is no or very little replication in the resistant line.

Line 316: The title and the distribution of virus in DSC260 described in the text (lines 319-321) do not match. Please correct. Further, please clarify whether it is epidermis or phellogen that is labeled in Figure 8B.

Line 360-370: Do you think that the chlorotic spots are a hypersensitive reaction? Then it is remarkable that only leaves show this reaction whereas roots show severe symptoms.

Line 366: What do you mean with “low efficiency”? Please indicate a percentage.

Figure 1b: Do you show here results from one plant or from the four plants? In the latter case, you should add error bars.

Figure 7C,D: The rectangle in C does not delineate the shape of D, so the closeup is not showing precisely the same region.

Figure 7F: The arrows are missing. Further, I am not convinced that the labeling in 7F is stronger than in 7B as you claim (text line 306). Maybe there is an error in the selection of the micrograph?

Figure 12Dp: I do not see any red label, but this could be due to the bad quality of the PDF file.

Figure 14C,D: please indicate the region chosen for the closeup.

Concerning terminology, I never read ‘phloemic’. ‘Phloem’ might do it. ‘Hystological’ should be replaced by ‘histological’ and ‘cotolydonary’ by ‘cotyledon’. I am not sure whether phellogen “cork cambium” is equivalent with epidermis. Please verify.

Author Response

Thank you very much for your critical reading and the important remarks to help to improve our manuscript.  We are grateful for all your efforts. Please find the improvements directly in the text and some responses to the questions raised.

Thank you!

You claim in the abstract that no CBSV resistance is known but then you present an immune variety (DSC167). This is somewhat contradictory. It seems that this is true for African cassava but not for South American varieties. Please clarify.

Abstract revised

Please add readable scale bars to all figures showing microscopic images.

Line 159: What is the peroxide concentration?

done

Line 264: Title 3.2 At best, you present indirect evidence that there is no or very little replication in the resistant line.

Title changed

Line 316: The title and the distribution of virus in DSC260 described in the text (lines 319-321) do not match. Please correct.

Upon CBSV infection by bud grafting, DSC 260 plants remained symptomless and there were no symptoms visible on all leaves, stems and tubers. In leaf sections subjected to RNAscope® ISH, the upper and the middle leaves were generally free of any signal. However, in the lower leaves, close to CBSV infected buds, red signals were occasionally found in abaxial and adaxial phloem cells (Figure 8, white arrows) and in cells of the spongy parenchyma (Figure 8A, red arrow).

Further, please clarify whether it is epidermis or phellogen that is labeled in Figure 8B. I am not sure whether phellogen “cork cambium” is equivalent with epidermis. Please verify.

The epidermis is a single cell layer followed by a cortex that contains three cortical layers;(phellem (cork), phellogen (cork cambium) and phelloderm (parenchyma cells). The innermost layer of the cortex is the endodermis, or the starch sheath in cassava, having starch storage function (Graciano-Ribeiro et al., 2009). Based on our signal localization the virus is mostly present in the phellogen cells)

Line 360-370: Do you think that the chlorotic spots are a hypersensitive reaction? Then it is remarkable that only leaves show this reaction whereas roots show severe symptoms.

We consider this as systemic hypersensitive reactions (SHR). The signal spreads along with the virus and more cells eventually become necrotized (discussed in lines 519-322). The symptoms in tuberous roots develop much later after several month of infection. In our opinion, this can be considered a hypersensitive response,

Line 366: What do you mean with “low efficiency”? Please indicate a percentage.

Figure 1b: Do you show here results from one plant or from the four plants? In the latter case, you should add error bars.

corrected

Figure 7C,D: The rectangle in C does not delineate the shape of D, so the closeup is not showing precisely the same region.

done

Figure 7F: The arrows are missing. Further, I am not convinced that the labeling in 7F is stronger than in 7B as you claim (text line 306). Maybe there is an error in the selection of the micrograph?

amended

Figure 12Dp: I do not see any red label, but this could be due to the bad quality of the PDF file.

done

Figure 14C,D: please indicate the region chosen for the closeup.

done

Concerning terminology, I never read ‘phloemic’. ‘Phloem’ might do it.

done

‘Hystological’ should be replaced by ‘histological’ and ‘cotolydonary’ by ‘cotyledon’.

Text revisions are done

Reviewer 3 Report

The manuscript deals with an issue of extreme importance, which is the tissue localization of cassava brown streak virus in resistant and susceptible lines. The authors studied by RNAscope in situ hybridization, the localization of CBSD on two germplasm lines and in different tissues of the plants.  The work is well executed and provides important information about the resistance of CBSV in cassava plants in order to develop future studies of genetic resistance.

The results are supported by evidence and my recommendation is to publish the manuscript.

Author Response

Thank you very much for critically reading our manuscript. We are very grateful for your positive response and support for research that will have a critical impact on cassava improvement for virus resistanc